# Generative and Evolutionary Techniques for the Process of Creating Architectural Objects on the Base of a 3D Prototype Model

Ewa Janina Grabska

Faculty of Physics, Astronomy, and Applied Computer Science, Institute of Applied Computer Science, Department of Design and Computer Graphics, The Jagiellonian University, 30-348 Kraków, Poland; ewa.grabska@uj.edu.pl

**Abstract:** The use of a genetic algorithm in evolutionary design is one of the major generative approaches for synthesis and evaluation during the design process. The process stimulates creativity in generating new, unexpected artifacts and aiding in their evaluation. We analyze the subject of the evolutionary design of building form styling following the aesthetic preferences of the designer. Component types and connection patterns characterize the building form and the rules of its composition. The designer using a graphics editor creates a 3D prototype model of a building form representative of his/her stylistic preferences by selecting different types of components and patterns of their connections. In the proposed evolutionary design, how the designer prototype model is organized, processed, and manipulated in generating buildings is based on the special graph structures. The research question addressed in this paper is, "How can a designer-defined 3D prototype model along with non-numerical graph calculations, influence computational creativity?" The main aim is to contribute to a better understanding of the non-numerical graph calculations describing the design process where visual perception is the driving force of creativity. Utilizing the developed formal description of the design synthesis, methodological contributions to generative and evolutionary techniques for computational creativity are presented.

**Keywords:** evolutionary design; computational creativity; visual perception; style; 3D prototype model; CP-graph; rewriting rules

## 1. Introduction

The subject of the paper concerns generative and evolutionary techniques from the perspective of supporting designers in the process of generating forms of buildings to the designer's stylistic preferences. By a style of buildings, we will understand a rule relating to a class of buildings for which its components can be structured by using the same set of aesthetic criteria [1]. Having to generate individual building styles is one of the biggest challenges in city generators. Buildings are procedurally difficult to generate due to their stylistic individuality, which is often the result of numerous architectural and cultural influences. Moreover, beauty perception is neither stable nor universal. Such an elaborate form of building styles for methods of Computer Aided Architectural Design (CAAD) needs an approximated or reduced model to limit the complexity. An attempt to solve this problem was presented in paper [2] for computer games. The main ideas from the paper are presented in a modified version here, in which we focus on the relation between the creativity of the designer and computational creativity in evolutionary design with the visual interactive environment, i.e., with the interface supported by visual modules such as graphics editor and visual communication module used during the design process. The designer creates a design object in a visual language using the graphics editor and receives system messages in this language, which can influence his/her creativity.

Research on the relation is considered here in the framework of computational creativity based on Computational Design Synthesis (CDS), which aims to support conceptual design by using formalization and a computer-aided process of finding solutions for design tasks [3,4]. The research carried out so far on the generation of stylized buildings has shown, on the one hand, that graphs are an effective tool representative for a variety of data, including architecture, urban, and planning designs, and on the other hand, that building style conceptualization is based on the visual and spatial perception. For this reason, the proposed CDS graphs are used to represent design objects created by the designer through a graphics editor. The process of the design synthesis is characterized by the variability of design structures modified by graph rewriting rules, i.e., the technique of creating a new graph out of an original graph algorithmically [5,6]. Within this framework, a stylized building is described in terms of the graph rule sequence and not as a static data block. The sequence of the rules reflects the sequence of the designer's actions in the visual environment. Possibilities expressing these actions as rules in evolutionary design have been explored by John Frazer using genetic algorithms since 1968 [7]. Evolutionary design is a generative method that has been used for many years to synthesize and evaluate designed objects during the design process [8,9]. Evolutionary design will be used to stimulate creativity in generating new, unexpected forms of building for evaluating the designer's aesthetic preferences.

The proposed approach combines CDS with evolutionary design from the point of view of an interactive visual environment, in which the designer creates *a 3D prototype model* to represent his/her stylistic preferences [10,11]. The designer's preferences are visualized as a configuration of distinct three-dimensional primitives attached. The analysis of the configuration enables a generation of buildings for which its components are structured using a set of the designer's criteria. Research to date has shown that significant relationships exist between prototypical and aesthetic evaluation [12]. It also turns out that the prototypical stimuli related to visual perception are processed faster and easier than non-prototypical stimuli [13]. Moreover, the basis of all empirical aesthetic measures is the construction of formal models of human perceptual processes [14]. This was one of the reasons that the conceptualization of stylistic preferences is here based on the Biederman visual perception model and assumes that the recognition of an object takes place through the exploration of three-dimensional structural components of the object, together with a description of the way they are connected [14]. An example of such an approach is the interactive system called Virtual City Creator (VCC) for computer games [2].

A central element of the presented design philosophy is the belief that the aesthetic experience of style depends on an individual's ability to visual perception in identifying a structural object [15]. Generally, only a person who has developed complex visual skills will be able to create a complex form of a 3D prototype model [16]. Figure 1 shows such an example of the prototype model of building style inspired by two built Eduardo Souto de Moura projects. It is known that visual communications concepts cannot be taught directly. Therefore, the methodology presented in this paper encourages the discovery of principles of visual design and not learning about them directly [17].

The knowledge of structured buildings in the proposed method is represented by composition graphs (CP-graphs) that define the relations not only between whole building components but also between fragments of these components at different levels of detail [18]. Design structure representations in the form of CP-graphs are particularly useful for creative design in engineering [19]. Since 2007, CP-graphs have been used in modeling the parallel direct solver algorithm utilized by the hp finite element method [20]. Research to date has shown that CP-graphs make it easier to describe designing both at the design object level and the design process level.

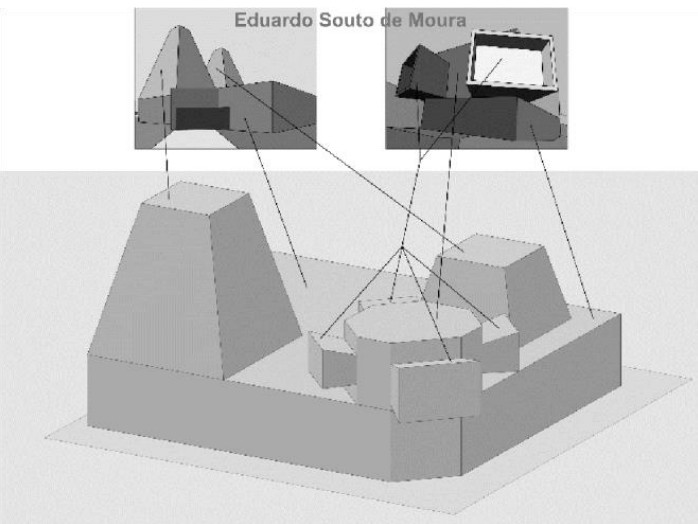

**Figure 1.** A prototype model inspired by two built Eduardo Souto de Moura projects.

Today, designers are faced with the need to recreate individual styles of buildings in areas such as computer games, movies, and commercials. There are generative methods for styling buildings, but the results are not satisfactory [21]. There is no generation tool for stylized building design commonly used to search for a new style during the design process. A shape grammar is the best-known generation tool. There are two types of shape grammar: analytical and original. Analytical shape grammar focuses on the one-time formalization of a particular architectural style. Original shape grammar focuses on the types of design rules that are generated and used during the design phase. However, no shape grammar can change the associated design space into a slightly changed version, whereas an actual architectural designer is capable of generating design alternatives that are beyond the design space [22].

The purpose of the work is to highlight the role of computational creativity with the use of generative and evolutionary techniques in the possibility of modifying design space during the design process. A better understanding of the design process based on human visual perception, which is the driving force of creativity, is necessary. Section 2 introduces CP-graph representations of building design structures, which are based on the Biederman model of human vision. Then, a system of CP-graph rewriting rules that describe the design process is presented. In Section 3, a constructive evolutionary procedure for styling buildings based on a visual metaphor reflecting the aesthetic requirements of the designer is described. In Section 4, computational creativity with an evolutionary design procedure based on non-numeric calculations, including visual perception, is characterized for aesthetic applications. It shows how computational creativity, along with the visual environment, can influence the designer's conceptual actions. The conclusion summarizes the research presented.

## 2. Innovative Designing and Composition Graphs (CP-Graphs)

By designed structure, we refer to the basic components and their relationships to a design object. Designing is innovation when dealing with the variability of the design structure [23]. A data structure based on graphs provides a fundamental principle to represent a design structure, while graph rewriting is an adequate description of design process actions.

### 2.1. Visualization and Graph Representation of Building Forms

In this paper, a 3D prototype model will be visualized as a set of three-dimensional generic primitives, together with a set of spatial connectivity relations among them. It is based on the qualitative volumetric representation proposed by Irving Biederman, both as

a model of human vision and a computer model in which 3D objects are constructed using three-dimensional structural components of objects called geons. Biederman developed the catalog of 36 geons that are classified by four qualitative features: edge (straight or curved), types of symmetry, size variation (constant, expanding), and axis (straight or curved) [14]. For instance, some geons in Figure 1 are truncated square pyramids; i.e., they are geons with the qualitative feature of size variation. These design objects represent so-called structural skeletons defined by their distinct parts attached. It is worth noting that in designing, it is always worth ensuring that the connection points between parts of objects are as clear as possible [24]. Two relations of attaching will be used between components of buildings: end-to-side and end-to-end.

In the proposed approach, the designer communicates with the design system using a visual design language on the monitor screen. A majority of visual languages are characterized by a vocabulary being a finite set of basic primitives and a finite set of rules specifying possible configurations of these. Visual languages often contain elements of vocabularies that have a precise meaning. It is impossible to discuss universal visual language because there are thousands of them in existence. In the first part of the creative design process, when designers attempt to express stylistic preferences regarding the form of buildings, they use visual language as an essential cognitive tool. The type and pattern of the form of buildings as well as their components are considered by the designer as stylistic criteria. They relate to the physical form and the rules of its composition [25]. The initial phase of designing combines some of the cognitive activities of normal seeing with the activities of visual imagination. Therefore, we propose a graphics editor based on Biederman's volumetric representation as a model of human vision. Designers begin with defining their vocabularies of the visual language by having a choice of four types of 3D objects with straight or curved edges, an appropriate kind of symmetry, size variations, and a straight or curved ax. They generate patterns for attaching appropriate elements of the vocabulary recalling, for example, existing architectural objects that serve as a reference or can be derived from metaphors. The use of the visual language makes the designer's ideas concrete and leads to the creation of his/her 3D prototype model.

To represent the design structures of such building descriptions using data, we need a specific type of graph that can describe the relations not only between whole building components but also between fragments of these components at different levels of detail. In the proposed method, these design structures are represented by CP-graphs that are especially useful for creative design in architecture [26]. A CP-graph is a labeled and attributed graph, where the components of the designed solid objects are represented by object nodes and bond nodes. The entire component is represented by the object node, while the bond nodes specify its parts. The labels of object nodes correspond to 3D-primitives for which their icons are also placed in the nodes.

Let us consider the 3D prototype model shown in Figure 2b; it consists of nine components, including the transformed 3D primitives in Figure 2a. The model in Figure 2b is represented by its CP-graph in Figure 3b. For each object node, the graphic code for any its bond is a small circle placed on the border of the node or of the bond (for hierarchical bonds). Bonds are numbered and represent the faces of solids or parts of the faces for hierarchical bonds. The number of bonds, without hierarchical bonds, in an object node (see: the node with label $a$ in Figure 3b) is equal to the face number of the cube represented by the object node (see: Figure 3a). The CP-graph in Figure 3b has an object node labeled $b$ with two hierarchical bonds. In this case, this node has seven bonds. Instead of the sixth bond, bonds 6.1 and 6.2 are considered, which represent parts of the 6th face of the solid with label $b$. Object nodes are equipped with two types of bonds: source bonds and target bonds; directed edges are drawn from source bonds to target bonds. The edges of CP-graphs are labeled. Each edge in Figure 3b corresponds to one of the two adjacency relations, named as either end-to-end or end-to-side. For the sake of simplicity, edges are drawn as a dashed line for the first label and a solid line for the second. Bonds that are

neither source nor target are called free, and they signal potential connections of their object nodes with other object nodes.

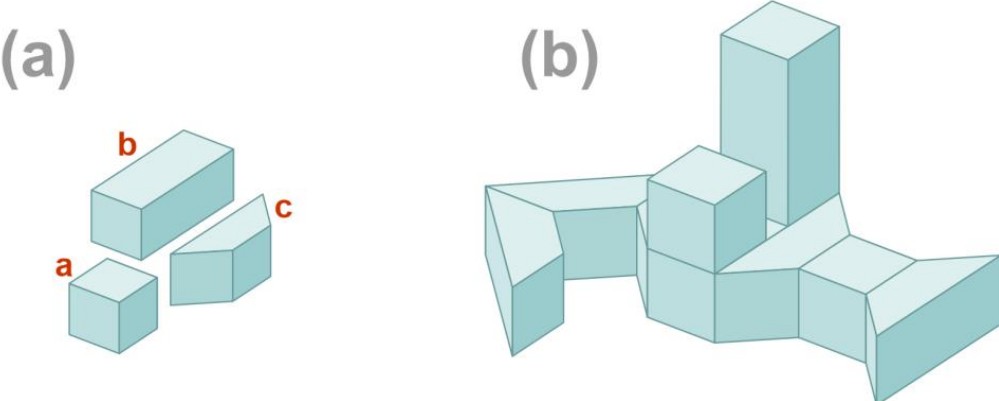

**Figure 2.** (**a**) Three basic primitives. (**b**) The 3D prototypical model.

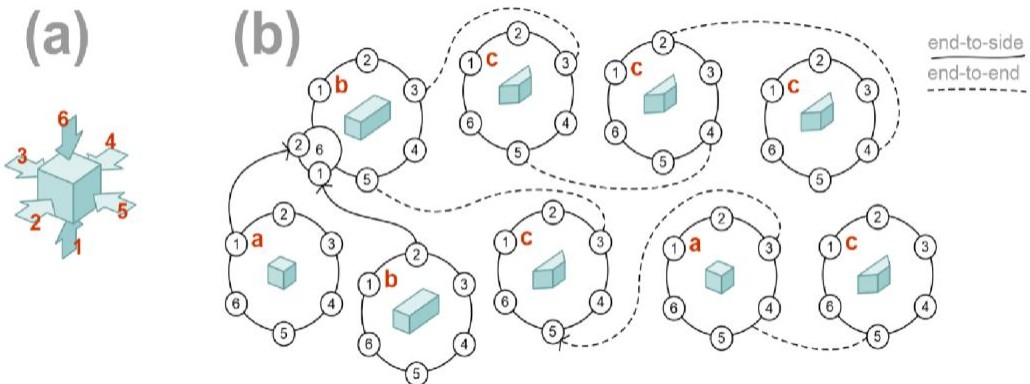

**Figure 3.** (**a**) The face numbering of a cube corresponds to the bond numbering in the object node representing the cube. (**b**) The CP-graph of the 3D prototype model in Figure 2a.

We formally start with the following definition of an attributed, labeled CP-graph with hierarchical bonds.

Let $\Sigma$ be an alphabet for labels of object nodes and edges. Let $A$ be a set of node and bond attributes and $P(A)$ be a set of all subsets of $A$. Let $P(B)$ be a set of all subsets of bonds from $B$.

**Definition 1.** *By an attributed, labeled composition graph (CP-graph) over $\Sigma$ and $A$, we mean a tuple $C = (V, E, B, ch, bd, s, t, lb, att_V, att_B)$, where we have the following*:

- *$V$, $E$, and $B$ are pairwise disjoint sets, for which their elements are called object nodes, bond nodes, and edges, respectively;*
- *ch: $B \rightarrow P(B)$ is a function that nests bond descendants such that none of a bond can be nested in two different bonds, and a bond cannot be its own descendant;*
- *bd: $V \rightarrow B^*$ is a function that specifies a sequence of different bonds for each object node, such that $\forall b \in B \ \exists! \ v \in V: b \in bd(v)$; i.e., each bond is assigned to exactly one object node;*
- *$s, t: E \rightarrow B$ are injective functions assigning to edges source and target bond nodes, respectively, in such a way that $\forall e \in E \ \exists v_1, v_2 \in V: s(e) \in bd(v_1) \wedge t(e) \in bd(v_2) \wedge v_1 \neq v_2$;*
- *lb: $E \cup V \rightarrow \Sigma$ is an edge and object node labeling function;*
- *$att_V: V \rightarrow P(A)$ is a function of object nodes attributes;*
- *$att_B: B \rightarrow P(A)$ is a function of bonds attributes.*

Let us denote elements of a CP-graph $C$ by $(V_C, E_C, B_C, bd_C, s_C, t_C, lb_C, att_{VC}, att_{BC})$. The CP-graph in Figure 3b describes only the structure of the building's form, i.e., its

components and relations between their faces. This is, however, not sufficient to represent a visual instance of such a building. This problem is solved by adding attributes to CPgraphs. In def. 1, the object nodes and bond nodes are attributed by means of the functions $att_V$ and $att_B$, respectively. The node attributes are divided into two groups: structural attributes, corresponding to properties used for defining component types, and descriptive attributes corresponding to metric properties that also define attributes for bonds. For CP-graph in Figure 3b, cuboid dimensions and the cube location data are examples of the metrical attributes for the object nodes with label *b* and for the 2nd bond of the 6th face of the object nodes with hierarchical bonds, respectively. Details of attribution are presented in [9]. It is worth noting that different numbers of attributes can be assigned to particular object nodes and bonds of CP-graphs.

Adding information about attribute values to a CP-graph *C* produces its graph instance. The CP-graph *C* is a basis for instances that differ only in the values of attributes. Formally, let $A_V$ and $A_B$ be sets of node attributes and bond attributes, respectively, and $A = A_V \cup A_B$.

**Definition 2.** *An instance of a CP-graph C over $\Sigma$ and A is a tuple I = (C, val_V, val_B), where we have the following*:

- $C = (V, E, B, ch, bd, s, t, lb, att_V, att_B)$ *is a CP-graph;*
- $val_V: V \times A_V \to D_V$ *is a function assigning attribute values to object nodes, where $D_V = \cup_{a \in AV} D_a$ is a family $D_V$ of sets $D_a$ of values of object node attributes, such that $\forall v \in V \; \forall a \in att_V(v)$: $val_V(v, a) \in D_a$;*
- $val_B: B \times A_B \to D_B$ *is a function assigning attribute values to bonds, where $D_B = \cup_{a \in AB} D_a$ is a family $D_B$ of sets $D_a$ of values of bond attributes, such that $\forall b \in B \; \forall a \in att_B(b): val_B(b, a) \in D_a$.*

In the following section, the instants of CP-graphs will be used to represent the design in the evolutionary design.

### 2.2. Design Process and CP-Graph Rewriting System

With advances in modern information technology, computers are used to generate innovative results. In this context, a data structure based on graphs provides a fundamental principle for representing a design object, while graph rewriting is an adequate description of the design process actions.

An innovative process of the design will refer to the generation of stylized buildings by the aesthetic preferences created by the designer in the form of visual data. As it has been considered, the designer's aesthetic preferences for style are presented as a 3D prototype model that is composed of distinct three-dimensional primitives attached.

A CP-graph of a 3D prototype model describes its design structure. However, further analysis is required to capture elements of building style. By a building style, we understand a rule relating to a class of buildings for which its components can be structured using the same set of aesthetic criteria [27]. These aesthetic criteria will be determined by the set of 3D primitives used by the designer as components and by a description of any possible way they are connected in the designer's creation of the prototype model. After reconstructing the structural properties of the form, a search is made for a *design space* containing all the buildings that meet the stylistic criteria specified by the designer. The primary tool for automatically creating a set of design solutions is a generative system that generates each solution by a sequence of rules. In our approach, buildings are represented by CP-graphs; therefore, CP-graph rule rewriting will be used to automatically generate derived buildings based on the prototype model.

In the proposed system (see: Figure 4) for the process of making architectural forms, the interface uses three modules: visualization, graphics, and control. A 3D prototype model created by the designer through the graphics editor consists of geons with appropriate qualitative features as components connected according to Biederman relations (end-to-end or end-to-side). The editor is integrated with the structure and rule analyzer, which defines an attributed CP-graph representation of the model and a set of CP-graph rules

for its generation. The new contribution is that different sequences of the elements of the CP-graph rule set using the Structure Generator enable the generation of different instances of CP-graphs, which are automatically transformed into new forms of buildings by the Visualization Module. Examples of new buildings are useful for the designer because provide helpful clues related to other possibilities of the use of patterns of connections than proposed in the 3D prototype model. In Figure 4, we have a path from the graphics module through the analyzer module, generator module, and visualization module. The designer using the new buildings can either modify the prototype model in the graphics module and repeat the cycle, which results in generating a new building proposal, or accept them by the control module as candidates for the initial genetic algorithm population of the building evolutionary generator.

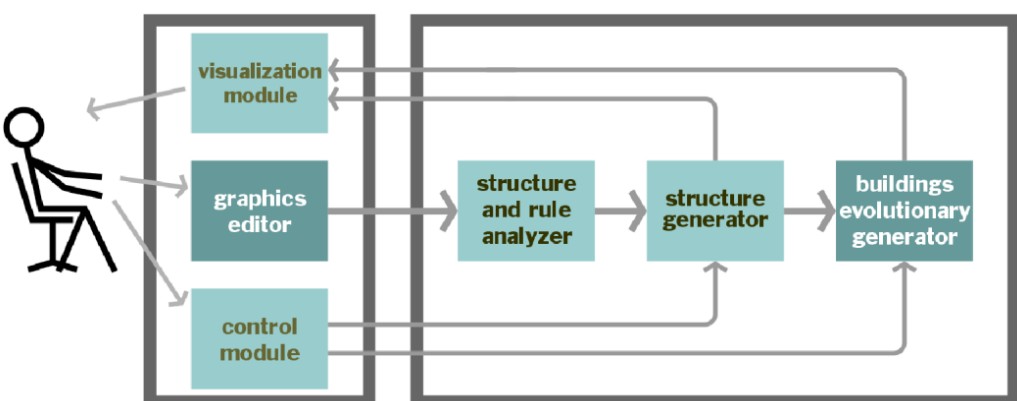

**Figure 4.** The main components of the system model for the generation of stylized buildings.

Examples of such buildings for the prototype model in Figure 2b with the set of CP-graph rules shown in Figure 5 will be discussed at the end of this subsection. The new buildings are accepted by the designer as elements of the initial population.

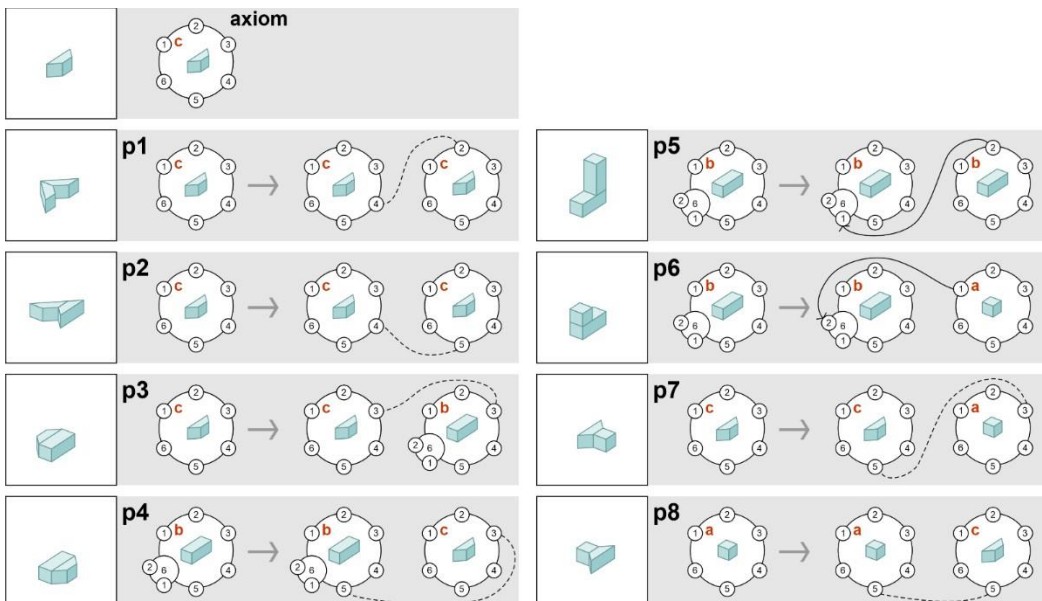

**Figure 5.** The system of the CP-graph rewriting rules for the 3D prototype model in Figure 2b.

Following the idea of the architect John Frazer for genetic algorithm, possibilities of expressing a designer's actions as generative rules are explored [7]. The designer creates a 3D prototype model by adding and attaching individual components. Figure 5 shows the steps of the designer during generating the prototype model and the eight CP-graph

rules that generate CP-graph (Figure 2b), recreating that process. This CP-graph rewriting system is context-free; that is, all left-hand-side rules are CP-graphs containing exactly one object node. The right-hand side of each rule contains an isomorphic object node to the left-hand side of the rule (informally, a copy of the left-hand side of the rule) and a new node is connected to it.

Let us denote by $\Gamma(\Sigma, A)$ the set of all attributed, labeled CP-graphs over $\Sigma$ and $A$, and by $\gamma(\Sigma, A)$, its subset that contains all CP-graphs with one-element sets of object nodes. Now, we are ready to formally define rewriting rules for CP-graphs.

**Definition 3.** *By a context-free CP-graph rewriting rules over $\Sigma$ and $A$, we understand a system $G = (P, s)$, where we have the following*:

1. *P is a finite set of rules $p = (l(p), r(p))$ satisfying the following conditions:*
   - *$l(p) \in \gamma(\Sigma, A)$. i.e., $l(p)$ is a CP-graph with a one-element set of object nodes $V_{l(p)} = \{w\}$, called the left-hand-side of p;*
   - *$r(p) \in \Gamma(\Sigma, A)$ and $\#V_{r(p)} = 2$ and $E_r \neq \varnothing$, i.e., $r(p)$ is a CP-graph with a two-element set of object nodes and with at least one edge, called the right-hand-side of p; and there exists $v \in V_{r(p)}$ such that $lb(v) = lb(w)$ and $bd(v) = bd(w)$;*
   - *$att_V(v) = att_V(w)$, i.e., the node v is isomorphic to the node w;*
2. *$s \in \gamma(\Sigma, A)$ is called the axiom of CP-graph.*

A context-free CP-graph rewriting rules are made for a generation. During the generation, the left-hand side of a rule is replaced by its right-hand side. Let us consider the system of the rewriting rules shown in Figure 4. We start with the axiom CP-graph containing one object node with label *c*. Each of the four rules *p*1, *p*2, *p*3, and *p*7 can be applied to the axiom graph. The CP-graphs, being the result of applying these rules, will be equal to their right-hand sides (formally, will be isomorphic to these right-hand sides). After applying one of the rules, we obtain a CP-graph containing two connected object nodes. If a rule is applied to a CP-graph with more than one object node, the necessary condition for the rule to be applied is the existence of an isomorphic object node in the CP-graph graph to the node of the left-hand side of the rule.

Let us assume that rule $p = (l(p), r(p))$ applies to CP-graph *C*. Applying *p* to *C* consists in moving the object node isomorphic to $l(p)$ from the *C* and inserting in its place the CP-graph $r(p)'$ that is isomorphic to $r(p)$. Then, the connections of the removed node are replaced with the connections of the object node of $r(p)'$ isomorphic to this node in such a way that each bond of the first is replaced with a bond of the second having the same ordinal number. As a result of applying production *p*, we obtain a new CP-graph *C'*. The relation between C and C' will be called the *direct derivation* of *C'* from *C* by the rule *p* and denoted as $C \Rightarrow^p C'$. The generation of the CP-graph in Figure 3b requires the use of all rules of the sequence in the CP-graph rewriting rules, shown in Figure 5, starting from the axiom. Such a sequence is called the *derivation* in the CP-graph rewriting rules, and it is denoted by $\Rightarrow^*$. Figure 6 presents the derivation of the CP-graph in Figure 2b. Since, formally, the symbol $\Rightarrow^*$ denotes a reflexive–transitive closure of the direct derivation relation $\Rightarrow$, it can be used as a designation for any derivation by applicable rule sequences in the rewriting rules system.

The derivation in Figure 6 is made in 8 steps using rule sequence *p*1, *p*2, ... , *p*8 and reflects 9 actions of the designer, as shown in Figure 7. The first action of the designer is to select a solid from an existing set of basic 3D primitives in the graphical editor or to create it. The solid is represented in the form of the object node and defined as an axiom of the CP-graph rewriting rules.

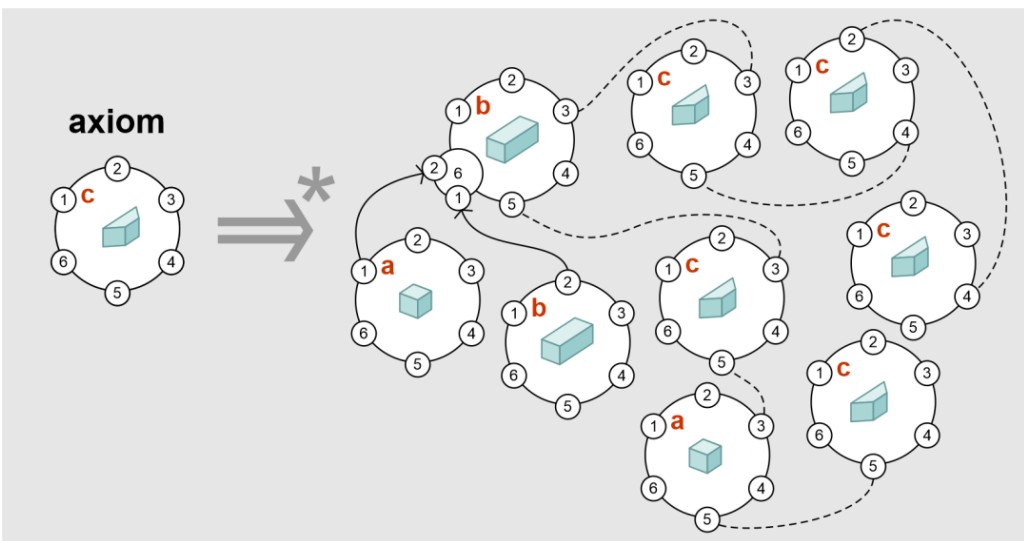

**Figure 6.** The derivation of the CP-graph in Figure 2b from the axiom.

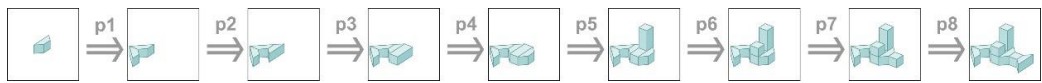

**Figure 7.** The actions of the designer are described by the sequence of rules.

When applying different sequences of the eight rules of the CP-graph rewriting rules in Figure 5, with the possibility of reusing the same rule, different CP-graphs are generated with the generator structure module (Figure 4). In the next step, the building creator module generates visualizations of the buildings corresponding to these CP-graphs by taking into account their attributes. Figure 8 shows examples of various buildings visualized from their CP-graphs generated using different sequences of these rules. The four buildings belong to the design space containing buildings compliant with the structural characteristics of the 3D prototype model in Figure 2b. Below these buildings are sequences of CP-graph rules from their derivations.

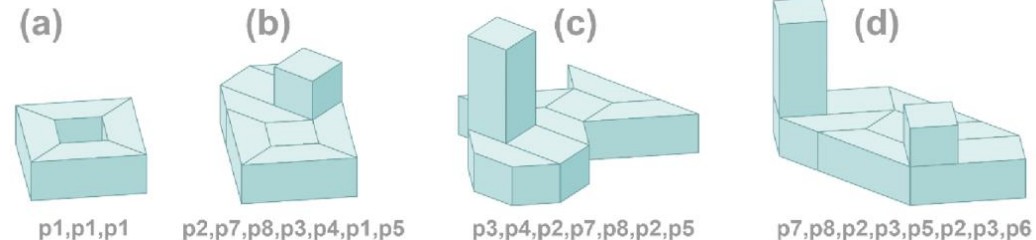

**Figure 8.** Elements of the design space containing all the buildings that meet the stylistic criteria specified by the designer in the form of the 3D prototype model in Figure 2b.

The urban complex presented in Figure 9 consists of buildings belonging to the design space under consideration. The following numerals in the figure read counterclockwise, starting with 3 on the right of the letter a, are the numbers of the rules applied.

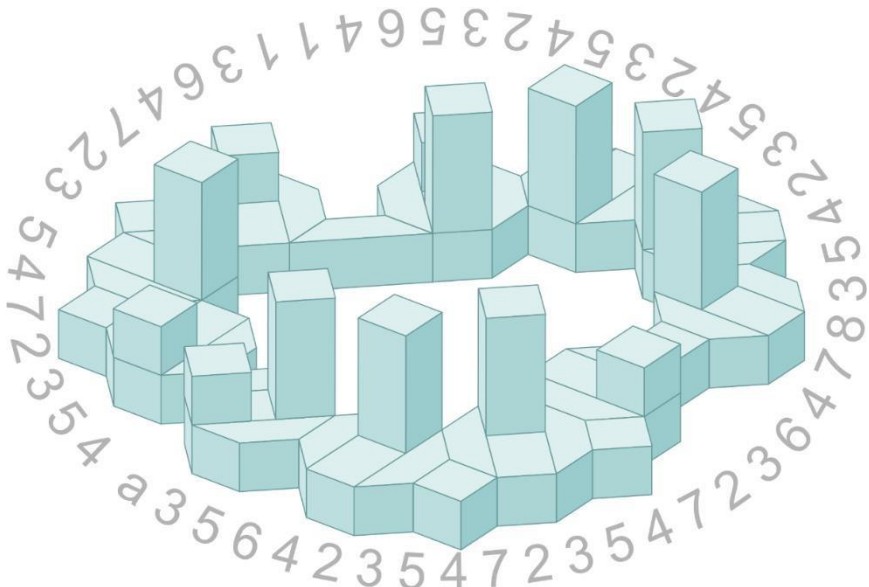

**Figure 9.** The urban complex.

## 3. Evolutionary Design and Computational Creativity

Computational creativity is about using a computer to generate results that would be regarded as creative if produced by humans alone [28]. In the proposed approach, the evolutionary design will be used to stimulate creativity in generating new, unexpected forms of building for evaluating their style. Evolutionary design is a generative method that has been used for many years to synthesize and evaluate designed objects during the design process [29].

In the Darwinian Theory, a phenotype is an individual's observable traits, while a genotype means the genetic contribution to the phenotype. At the design level, the phenotype will be understood as a description of the designed object with the possibility of visualization, while the genotype will be its representation enabling structural and attribute changes. The fitness function in designs uses the evolutionary theory of the improvement of the organism's performance by striving for excellence in creating artifacts. This is performed by improving the designed object at the genotype level and outside of the designers without too much effort on their part. The design process simulates the evolutionary process of crossover and mutation at the genotype level [30].

In our approach, a genotype of the building will be represented by a sequence of CP-graph rules used for the derivation of its CP-graph structure, while a phenotype will be the visualization of the three-dimensional structured building, i.e., the configuration of its 3D components.

### 3.1. Initial Population and Genetic Operators

The process of evolutionary design starts with an initial population of individuals with the required design characteristics represented by a sequence of genotypes corresponding to the individuals. Most evolutionary design systems generate new artifacts based on random initial populations. It is recommended that the population size be less than ten because then there is a rapid evaluation in each generation [31].

In this paper, the initial population contains genotypes for buildings for which their structures are generated by CP-graph rules from the set of rules of the derivation of CP-graph representing the design structure of the 3D prototype model. Additionally, it is assumed that for each rule from this set of CP-graph rules, there exists at least one genotype in the initial population in which this rule is applied to derive this genotype.

Since the evolutionary design process takes place at the genotype level, where genotypes are represented by CP-graph rule sequences, genetic operators for such representation

should be defined. A mutation operator defined on one genotype is used to introduce new properties to the population, both at the structural and visual levels. In this paper, the mutation operator can modify the structure of the building by removing or adding a CP-graph rule or changing the value of node attributes. The crossover is performed on two selected sequences of CP-graph rules representing the genotypes. The crossover operator requires establishing one rule from each sequence that would be exchanged during the evolution process.

Examples of the use of genetic operators in creating new forms of buildings will be illustrated with the 3D prototype model and its CP-graph shown in Figures 2b and 3b, respectively. The initial population will be the four building forms shown in Figure 8, their phenotypes, and four sequences of CP-graph rules numbers representing their genotypes. All these CP-graph rules are presented in Figure 5.

Consider the mutation operator applied to the building genotype shown in Figure 8c as a CP-graph rule sequence. The operator will add a new CP-graph rule to the rule sequence from which the structure of this building is generated. Only the primitives shown in Figure 2a are used to create new CP-graph rules, while new connections can be defined between the free bonds of object nodes appearing in the CP-graph in Figure 3b, which defines the structure of the prototype model shown in Figure 2b. As it has been considered, the free bonds of a CP-graph represent its potential connections. In that case, it is a natural extension of the CP-graph structure of the prototype model. This type of CP-graph rule will be called a structurally admissible rule.

Figure 10 shows the considered building at the phenotype and genotype levels, the addition of a structurally admissible rule, and the result of applying the mutation operator.

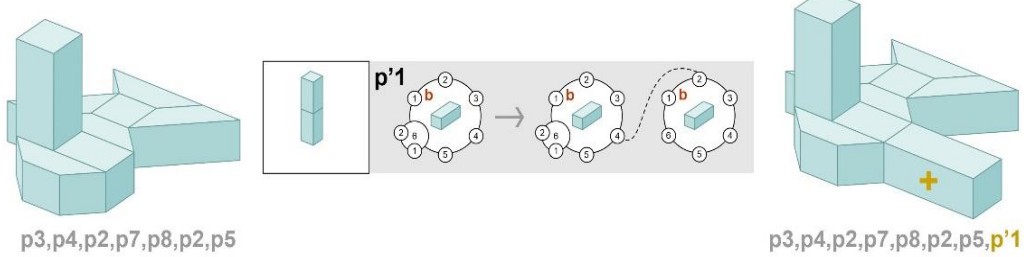

**Figure 10.** An example of the use of the mutation operator by adding a structurally admissible rule at phenotype and genotype levels.

Figure 11 shows another example of applying the mutation operator to the genotype of the building in Figure 8d. This time, the CP-graph rule is removed.

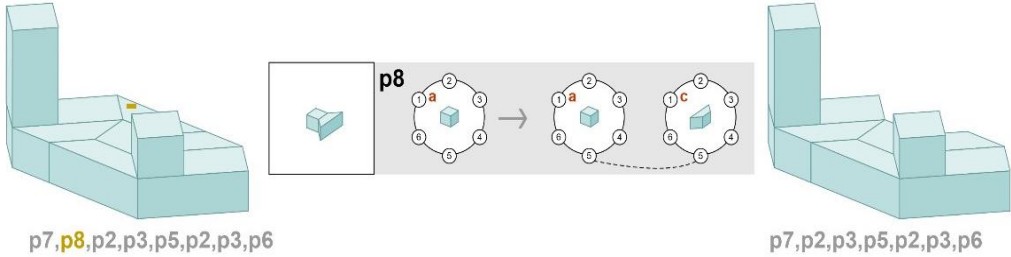

**Figure 11.** Another example of applying the mutation operator at phenotype and genotype levels.

Let us consider a crossover operator for which its two arguments are sequences of CP-graph rules representing genotypes. Let us select the two buildings as arguments after applying operator mutations in Figures 10 and 11. In this case, new offspring are first generated by mutating the parent copies. Figure 12a shows the arguments of the crossover operator and rules p2 and p5 (see: Figure 5) at the genotype level for changing the building genotypes. Figure 12b presents the results of the crossover operator.

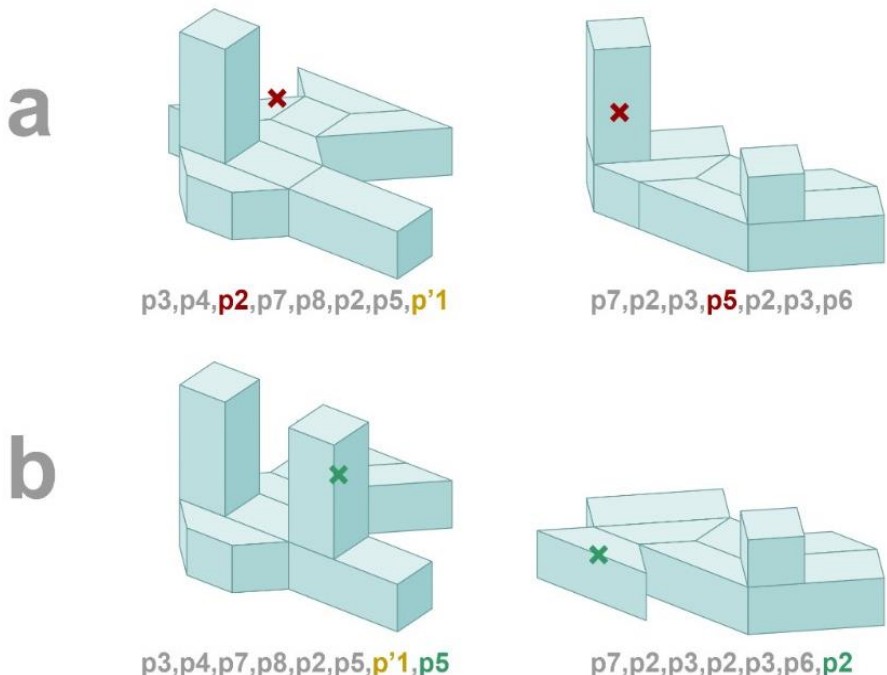

**Figure 12.** An example of applying the crossover operator at the phenotype and genotype levels: (**a**) arguments of the crossover operator; (**b**) the results of applying this operator.

The following condition must be satisfied to use the crossover operator. After crossover is applied, the design structures of the operator's results at the phenotype level must be connected to CP-graphs in a topological sense; i.e., in each of these CP-graphs, there is a path from any object node to any other object node.

Enabling the addition of structurally admissible CP-graph rules shown in Figure 13 by the use of mutation and crossing operators for the presented structures can lead to interesting results presented in Figures 14 and 15.

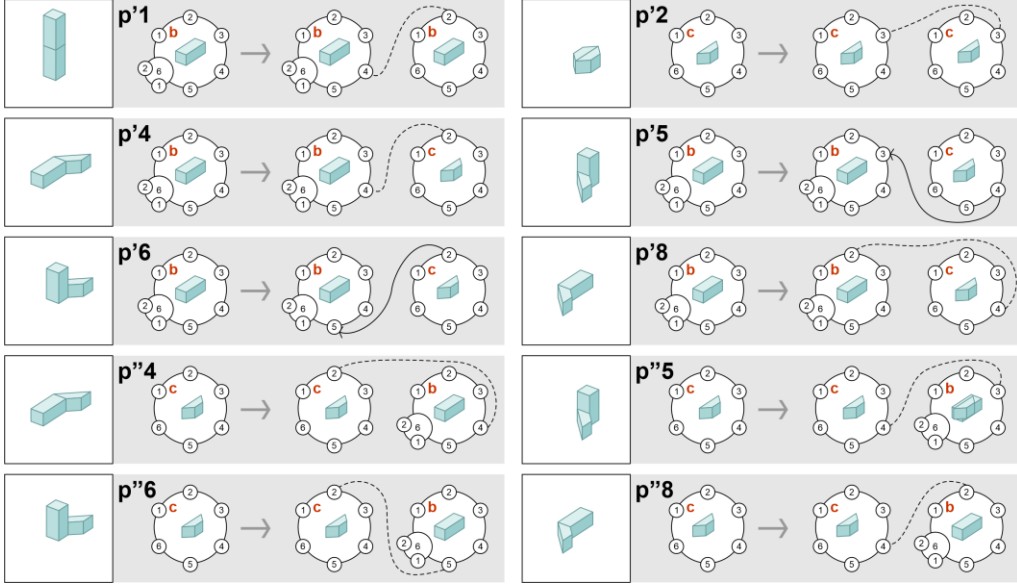

**Figure 13.** Examples of structural admissible CP-graph rules for the CP-graph in Figure 3b.

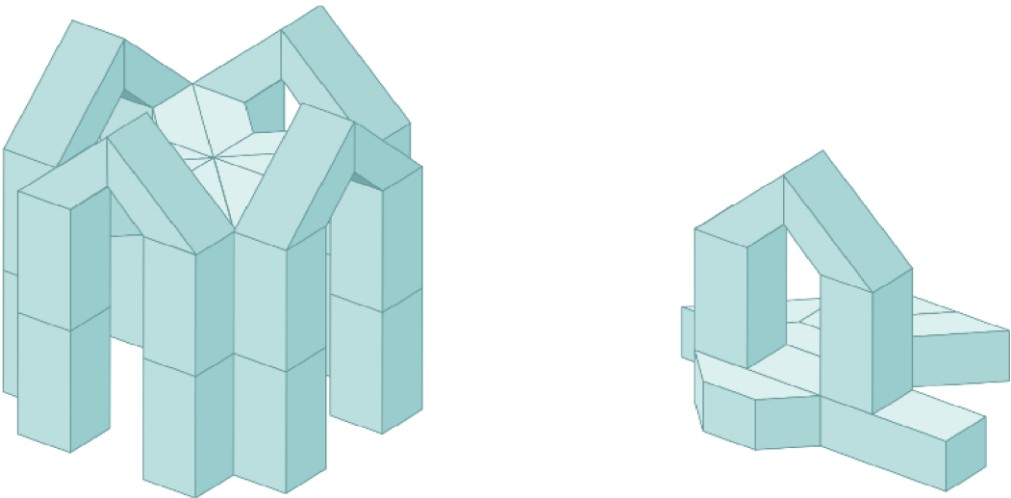

**Figure 14.** Examples of stylized buildings.

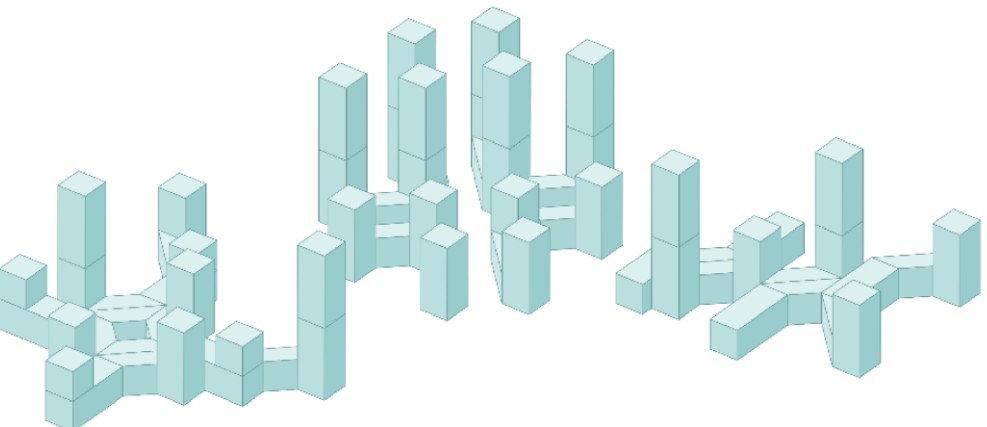

**Figure 15.** Other examples of stylized buildings with admissible relations.

*3.2. Evaluation*

In evolutionary designs, the fitness function plays a fundamental role in the selection process for reproduction, i.e., in selecting the best individuals as arguments for mutation and crossover operators. In the subject under investigation, which buildings will be selected as the best fit depends on what architectural aspects are included in the definition of the fitness function.

The computational aesthetic evaluation for a given class of objects utilizes software applications of aesthetic measurements. The previously proposed methods of aesthetic evaluation for buildings were based on Birkhoff's aesthetic measure adapted for 3D solids and on the Biederman visual perception model [32,33]. There exist examples where the fitness function is determined by fuzzy evaluation of designs, which determines to what degree each phenotype fulfills the aesthetic criteria [9].

In our approach, a two-stage fitness function needs to be developed. An interesting example of such an approach is presented in the DARCI system by simulating a virtual image artist [15]. In the presented method, the first step should be a function that determines the degree of structural similarity between the generated buildings and the structure of the prototypical 3D form. In the assessment of character similarity between two buildings, the factors such as the number of building components and number of component types are provided by the graph rewriting rules for the derivation of the CP-graph, which represents the design structure of the 3D prototype model. This fitness feature evaluates the degree of the style's structural similarity. In the second step, some aesthetic qualities of the generated buildings should be considered. Since visual perception is a decisive factor in the aesthetic

evaluation of building forms, this evaluation usually uses the human visual perception model. Buildings are seen as configurations of some basic solids, which are based on the Recognition-By-Components (RBC) theory developed by Biederman. The analysis of the properties of components of the generated buildings and the relationships between them is a basis for evaluating the elements of beauty, such as, for example, order, harmony, and rhythm [34].

After many attempts at the esthetic evaluation of architectural objects with the use of a computer, there is also the opinion that the aesthetic value of artifacts is related to their ability to provide recipients with stimuli enriching their aesthetic experiences. Research on a model capable of reproducing the mind's reaction is much more challenging than generating artifacts [35]. In the presented approach, one of the most important factors of the automatic aesthetic evaluation based on the prototype model created by a designer is the use of the aesthetic assessment of the representativeness of each generated building in the category meeting the designer's preferences [36]. Therefore, during the evolutionary process, the fitness function to evaluate aesthetics should be performed not only by the software but also by the designer [37,38].

## 4. Discussion

Contemporary research into the way designers work suggests that while the initial idea may be a cognitive act, the main work of creative design is through a kind of dialogue with a computer interface often supported by computational creativity. In this section, we will discuss how computational creativity using the proposed generative and evolutionary techniques allows the designer to modify the design space during the process of designing new artifacts based on aesthetic criteria.

The mechanism of computational creativity is known to exist regardless of any particular field. We examine this mechanism using non-numerical calculations based on visual perception in the field of creating stylized buildings. From this, we attempt to characterize computational creativity with the use of some concepts of the *computational ontology*. From a computer science point of view, the basic definition of ontology is as follows: "An ontology is an *explicit specification* of a conceptualization". In other words, ontology is an analysis of relevant entities and an organization of them into concepts and relations [39].

In the first step of the conceptual design process, we are dealing with the creativity of a designer who uses visual tools to propose aesthetic experiences into style, expressing them through a 3D prototype model. It is an externalization of the conceptualization from the mind of the designer. During this process, the designer selects the 3D primitives that are components of the prototype model and decides how to connect them. In this way, the designer determines a visual language for which its vocabulary is a finite set of primitives, and a finite set of rules specifies the possible configurations of these primitives. We say in this case that the visual language is *committed to* the designer's conceptualization in the interactive environment.

From the ontological point of view, representing knowledge formally is based on the designer's conceptualization: the attributed objects that are assumed to exist in some area of interest and relations among them. Apart from the externalization of the designer conceptualization, we need its specification explicitly as input data for the system. In the proposed approach, the conceptualization is specified on the basis of the designer's externalization that can be automatically transformed into an appropriate attributed CP-graph. There exists an ontological commitment between attributed CP-graph language as data for the system and the externalization of the designer's conceptualization in the form of the visual language. It is worth noting that the CP-graph representing the prototype model, due to its free bonds, provides the potential to introduce structural admissible rules, i.e., CP-graph rules with new possible relations between components, which means CP-graphs represent an extensional type structure. The phases of obtaining the specification of conceptualization explicitly with the use of a visual environment and using it in the evolutionary design process are presented in Figure 16.

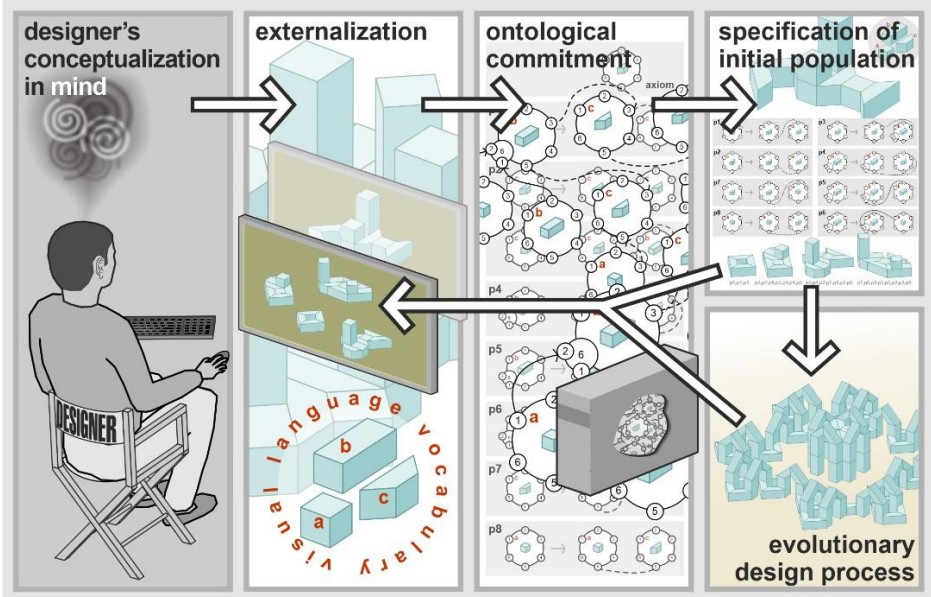

**Figure 16.** Transformation of designer's conceptualization into its specification.

The rule rewriting system based on a CP-graph allows the reflection of the designer's actions in the computational creativity system.

To discuss the creative evolutionary mechanism, we will use an abstract mental schema that acts as a basic structure for arranging ideas about computational creativity.

The scheme consists of the following seven elements:

1. Procedure selection;
2. Ingredient selection;
3. Lower bound definition;
4. Upper bound definition;
5. Construction;
6. Selection;
7. Reflection.

From a procedural point of view, the proposed conceptual concept scheme can apply to most computer-supported artifacts generating processes as well as to those created by humans [35]. We start with the selection of a constructive procedure, and then we select the elements that ensure the operation of the constructive procedure. In the proposed approach, the mutation and crossover operators make up a constructive procedure. Mutation and crossover operators are developed to increase the structural diversity of buildings. Initial elements to feed the procedure are presented in Figure 17c,d.

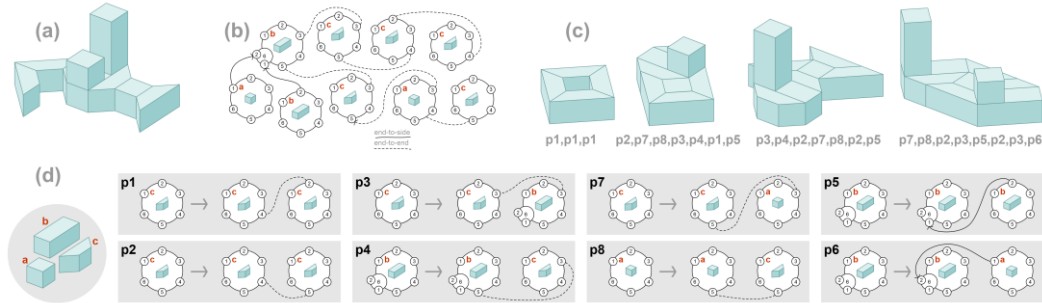

**Figure 17.** Elements to feed the constructive procedure obtained on the base of the 3D prototype model and its CP-graph in (**a**,**b**), respectively; (**c**) the initial population; (**d**) principles specifying connections between primitives.

The initial visual language for the constructive procedure at the phenotype level is included in the initial population. The language consists of the three-element vocabulary of primitives and a finite set of principles specifying connections between these primitives. These principles are contained in CP-graph rules (Figure 17d). It is assumed that each CP-graph rewriting rule in the derivation of the CP-graph for the prototype model will appear at least once in the representation of some genotype of the initial population. The CP-graph rewriting rules presented in Figure 17d implicitly define the structures of buildings that meet the stylistic criteria determined by the designer.

From a computational creativity point of view, we are looking for similarities between the actions of the designer during the design process and the operations undertaken in the creative system. There are four types of actions that a designer performs: physical actions such as drawing and erasing design elements, conceptual actions where new design goals and requirements are defined, perceptual actions, and functional actions, e.g., design evaluation. A mutation operator reflects the physical actions by adding and removing CP-graph rules. However, a crossover operator recreates the two types of the designer's actions applied to the earlier conceptions of solutions: first perceptual actions and then the physical actions aimed at crossover a pair of CP-graph rules in these solutions. A designer's conceptual actions, in which new design goals and requirements are set, can have a significant impact on modifying the original design space. In the considered evolutionary constructive procedure, the reflection of conceptual actions of the designer will be the application of structurally admissible CP-graph rules in genetic operators (see Figures 10 and 13).

The third and fourth elements of the conceptual scheme, the lower bound definition, and the upper bound definition refer to the determination establishing the candidate rejection criteria and the candidate success criteria, respectively. In the former case, the goal of the fitness function is to prevent a complete change of the designer-defined style with the 3D prototype model. In the considered method, the fitness function can enumerate the degree of compliance with the design requirements by a phenotype about the 3D prototype model at the genotype level, e.g., assuming that the ratio of the number of applying structurally admissible CP-graph rules to the number of applying all the CP-graph rules in the derivation of the genotype must be less than 0.5. In the latter case, taking into account that the style of the building is determined by the designer, first of all, the aesthetic criteria included in the prototype of the three-dimensional model based on the Biederman model should be recommended. Examples of such aesthetic measures are presented in [9,32]. The calculation of the fitness function values reflects the functional action taken by the designer.

Summarizing the first five elements of the abstract mental scheme in the considered scheme evolutionary constructive procedure, we can find a reflection of all four types of designer actions, i.e., physical, perceptual, conceptual, and functional. The proposed non-numerical graph computations with an evolutionary mechanism, together with the visual environment, constitute adequate mechanisms for the development of computational creativity.

Selection, the sixth element of the scheme, is the application of the rejection and success criteria to the results of the constructive procedure. It is a process for determining the selected phenotypes according to their fitness. The selection schema may force a process toward divergence or convergence [12]. To retain a balance between the designer's creativity in defining a style of buildings and new forms of buildings proposed by computational creativity, a visual environment should be designed to help the designer in evaluating building fitness and select them for the next cycle of the processor to terminate the system.

## 5. Conclusions

In the presented approach, the designer provides information about a dedicated visual language, when creating a prototypical 3D form using a graphics editor, which is an introduction to the search for a design space for stylized buildings consistent with aesthetic preferences. Elements of this language transformed into CP-graphs with free

bonds provide extensional structures for the system of the graph rewriting rules that enable the generation of elements of the design space. Some of these elements are input data relative to a constructive evolutionary mechanism, the results of which can modify the design space during the design process. This possibility gives the architectural designer greater freedom to create design alternatives that extend beyond the original design space.

It should be noted that in the case of more stylistically complex forms of buildings, the proposed graphics editor will need to be modified. We are planning a possibility to create a 3D prototype model from several of its prototype sub-models, also consisting of attached 3D solids, with each sub-model reflecting different aspects of the style defined by the designer. The CP-graph structure of such a 3D prototype model will be generated by a system consisting of rewriting systems of its sub-models. The complex system of CP-graph generation based on CP-graph subsystems has been developed and implemented for the finite element method [40].

It is assumed that the designer creates his 3D-prototype model by adding one three-dimensional shape at a time. The designer's synthesis process consists of a sequence of actions determined by micro-evaluations of a defined style. The designer evaluates both the types of components and how they are connected. The case will be simplified when the configuration of several components is repeated in the prototype model. Components of such a configuration will be aesthetically analyzed by the designer only during the first time and then the configuration will be treated in the same manner as the other single components; i.e., it will be added to the visual language vocabulary. Consequently, the system of the CP-graph rewriting rules for the 3D prototype model with the repeating configuration of components will have to include two new rules, one replacing the configuration with a single CP-graph node and another reverse. The formalism of CP-graphs provides such a transformation. The CP-graph corresponding to the configuration can be seen as a CP-graph node if only its external bonds are defined.

Our approach proposes the theory and developing methods of communicating with a computer for non-numerical computations on applications such as information retrieval, design generation, and design evolution. We would like to inspire designers, especially architects, to search for interesting computer-aided research opportunities that are not only closely related to engineering numerical computations.

In the future, a framework for unsupervised CP-graph representation learning will be explored to classify and detect the degree to which a building is representative of a building class [41]. Unsupervised artificial intelligence techniques can be useful in determining the typological and topological features of architectural solutions and we hope that they will also be helpful in aesthetic evaluations.

**Funding:** This research received no external funding.

**Institutional Review Board Statement:** Not applicable.

**Informed Consent Statement:** Not applicable.

**Data Availability Statement:** The study did not report any data.

**Conflicts of Interest:** The author declares no conflict of interest.

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
