# Peer review of "Generative and Evolutionary Techniques for the Process of Creating Architectural Objects on the Base of a 3D Prototype Model"

_buildings, doi:10.3390/buildings12070899_

Round 1

Reviewer 1 Report

Overall the paper figures are well developed and the topic is an interesting one. I am less clear what the specific novel contribution is as the 3d prototype based approach remains unclear to me after reading the paper - also how those prototypes are defined by a user to start the design process would be important to understand in order to capture human creativity.

The figures are overall well prepared and clearly readable. But part of the challenge would be how to interface this underlying structures to the human designers for manipulation which is not addressed other than abstractly in figure 3.

The notation and formal definitions of the process seem thorough and well developed.

Was the approach ever tested with human designers other than the author? If yes it would be helpful to include some form of case study of how the proposed approach helped expand the design space for human designers. Especially with respect to the formulation of a 3d Prototype building as a starting point.

Detailed comments - the overall paper needs thorough editing - I am providing some examples and suggestions here - but there is more

Abstract
Unsure what “genetic technology” means here – I am familiar with Genetic algorithms in design – but genetic technology I would see positioned in biotech and related to hardware rather than software or design abstractions? It is used once more in relation to Frazer so likely a term he used as well but am still unconvinced by it.

Line 7 “during design process” -> “during the design process”

Line 8 “generating a new, unexpected artifacts and” -> “generating a new, unexpected artifact”

Line 10 “prototypicality” -> “prototypical” why a noun?

Line 14 “this paper addresses” – the paper is not a person and cannot address anything – you as the author address the question – also “the paper” is unnecessary to mention and is used too often throughout

Line 15 “This paper aims at contributing to a better understanding” – same remove

Line 18 “the paper provides” - same

Line 24 “the paper deals” – same remove
Line 25 “from the perspective of supporting designer’s” -> “from the perspective of supporting designers”

Line 34 “but the aim of this paper” -> same – remove with a different formulation without “the paper”

Line 35 “concerns the relations between” -> “concerns the relation between”

Line 36 “with the visual interactive environment” – unclear what this refers to  - is the visual interactive environment what creates the relation between creativity of the designer and computational creativity? Please edit to clarify

Line 36 “This task here is considered in the framework of” ->”this task is considered here in the framework”

Line 42 “architecture, urban and planning designs” -> “architecture, urban, and planning designs”

Line 44 “this paper uses” – to frequent use of this paper- and again the paper is not using anything, but the researchers are.

Line 53 “In this paper” – again unnecessary – everything here other than cited work is in this paper

Line 56 “presented in this paper” ->remove, unnecessary

Line 72 “An example such an approach” -> “As an example of such an approach”

Line 75 “on an individual's ability to perceive visual in identifying a structural object” -> “on an individual's ability to perceive style(??) visually in identifying a structural object”

Line 97 “But any shape grammar has not an ability to change the associated design space, into a slightly changed its version” -> please edit – sentence unclear – do you mean “no shape grammar has the ability to change the associated space into a slightly different version?

Line 103 “Moreover, this paper aims at contributing” -> the paper does not aim – the researchers have aims and write a paper – but not necessary to state this its wordy- I would write something like “A better understanding of the design process….is necessary”

Line 106 – “this paper” again unnecessary -remove -simply start “Section 2 introduces…

Line 218 “computers is used to generate” -> computers are used to generate

Line 219 “context, a data structure based on graphs provide a fundamental” -> provides

Line 222 “In the paper” -> remove

Line 226 “designer creates a prototypical using a”-> “designer creates a prototype (or- a prototypical design?) using a”

Figure 3 – it is unclear how the designer does any of this with regards to interface – what is the design of the graphic editor – how does the control module work and how are those elements merged?

Line 420 “In our approach, two-stage should be developed” -> “In our approach, a two-stage needs to be developed(?)”

Line 424 “In assessment of character similarity” -> “In the assessment of character similarities”

Line 425 “factors such as: a number of building components and a number of component types” -> “factors such as:  number of building components and number of component types”

Line 427 “This fitness feature could protect with a complete change of style.” – sentence does not make sense please rewrite – what is protecting what?

Line 430 “forms, this evaluation usually use the human visual perception models” -> “forms, this evaluation usually uses the human visual perception model”

Line 431 “In this paper” remove

Line 436 “On the other hand” – it is not possible to use on the other hand without On the one hand first – please use an alternative

Line 437 “one should agree with the opinion” – unclear – why should one agree?

Line 443 during evolutionary process -> during the evolutionary process

Line 451 “ This paper examines” – remove – a paper does not do anything -the authors do – the paper contains – but unnecessary – We examine –

Line 453 “creativity with the use some concepts of computational” -> “creativity with the use of some concepts of computational”

Line 455 “ the ontology analyses relevant entities and organizes them into concepts and relations” – “the ontology is an analysis of relevant entities and an organization of concepts and relations

Line 457 “In the first step of conceptual design process,” -> “In the first step of the conceptual design process,”

Line 464 “that the visual language commits to the designer’s conceptualization in the interactive environment. “ -> unclear – a visual language cannot commit to anything – please edit and clarify -the visual language is committed?

Line 472 “The attributed CP-graph language is another language commits to a conceptualization 472 representing basic knowledge for the system.” – sentence incomplete – does not make sense what does “is another language commits” mean?

Line 513 “From computational creativity point of view, we are”-> “From a computational creativity point of view, we are”

Line 516 “such as drawing and erasing elements design” -> “such as drawing and erasing design elements”

Line 518 “A mutation operator by adding and removing CP-graph rules reflects physical 518 designer’s actions. -> “A mutation operator reflects the actions of a physical designer by adding and removing CP-graph rules.”

Line 524 “designer will be application of structural admissible CP-graph rules” ->” designer will be the application of structural admissible CP-graph rules”

Line 529 “In the former case the goal of fitness function is” -> “In the former case the goal of the fitness function is”

Line 544 “The proposed non numerical graph computation with an evolutionary mechanism, together with the visual 545 environment, constitute, in accordance with the definition of computational creativity, ad-546 equate mechanisms for its development.” – this sentence is hard to follow and should be edited to clarify its content

Line 557 “In the presented approach, the designer, when creating a prototypical 3D-form using a graphic editor, provides information about a dedicated visual language,” -> “In the presented approach, the designer, provides information about a dedicated visual language, when creating a prototypical 3D-form using a graphic editor,”

Line 566 “This paper proposes the theory” – remove paper cannot propose anything

Line 568 “It is hoped that this paper “ – remove It is hoped is not possible grammatically- remove this paper…

Line 574 “and it is hoped that they” – “we hope”

Reviewer 2 Report

The paper demonstrate how state of the art techniques (CDS with evolutionary design) may be combined using the concept of prototypicality to characterize the preferred style of a class of buildings.
A workflow is described and discussed. Althought shape grammar and evolutionary design are not new, their usage to provide a 3D-visual metaphor of style applying non-numerical graph calculations is original and useful.

Round 2

Reviewer 1 Report

Thank you for the responses to the comments and changes. I am still unconvinced though by the lack of clarity on how a designer will specify the style through an interface and figure 16 has not improved to clarify and neither has figure 4. Both still leave out the crucial step of how a design idea is codified into a externalization and how the quite limited block language can succeed in capturing a range of architectural styles. The examples given ok in their limited range but not representative of the flexibility needed to capture different styles. The lack of user tests with designers is also a short coming that does make me less convinced of the approach.

Round 3

Reviewer 1 Report

Thank you for expanding further on the interface for design system.
